# Terrestrial Animal Source Foods and Health Outcomes for Those with Special Nutrient Needs in the Life Course

**DOI:** 10.3390/nu16193231

**Published:** 2024-09-24

**Authors:** Lora Iannotti, Ana María Rueda García, Giulia Palma, Fanette Fontaine, Beate Scherf, Lynnette M. Neufeld, Rachel Zimmerman, Patrizia Fracassi

**Affiliations:** 1E3 Nutrition Lab, Brown School, Washington University, St. Louis, MO 63130, USA; r.b.zimmerman@wustl.edu; 2Food and Agriculture Organization of the United Nations, 00153 Rome, Italy; ana.ruedagarcia@fao.org (A.M.R.G.); giulia.palma@fao.org (G.P.); fanette.fontaine@yahoo.fr (F.F.); beate.scherf@fao.org (B.S.); lynnette.neufeld@fao.org (L.M.N.); patrizia.fracassi@fao.org (P.F.)

**Keywords:** terrestrial animal source foods, nutrition in the life course, microbiome, food allergies, pregnancy and lactation, young children, adolescents, older adults

## Abstract

**Background.** Animal source foods are under scrutiny for their role in human health, yet some nutritionally vulnerable populations are largely absent from consideration. **Methods.** Applying a Population Intervention/Exposure Comparator Outcome (PICO/PECO) framework and prioritizing systematic review and meta-analyses, we reviewed the literature on terrestrial animal source foods (TASFs) and human health, by life course phase. **Results.** There were consistent findings for milk and dairy products on positive health outcomes during pregnancy and lactation, childhood, and among older adults. Eggs were found to promote early childhood growth, depending on context. Unprocessed meat consumption was associated with a reduced risk for anemia during pregnancy, improved cognition among school-age children, and muscle health in older adults. Milk and eggs represent a risk for food sensitivities/allergies, though prevalence is low, and individuals tend to outgrow the allergies. TASFs affect the human microbiome and associated metabolites with both positive and negative health repercussions, varying by type and quantity. **Conclusions.** There were substantial gaps in the evidence base for studies limiting our review, specifically for studies in populations outside high-income countries and for several TASF types (pig, poultry, less common livestock species, wild animals, and insects). Nonetheless, sufficient evidence supports an important role for TASFs in health during certain periods of the life course.

## 1. Introduction

### 1.1. Terrestrial Animal Source Foods (TASFs)

Terrestrial animal source foods (TASFs) are targeted in our contemporary discourse around global trends in human health and the environment (Box 1). As is often the case in a contentious debate, certain perspectives and subtleties may be lost particularly for those of more vulnerable individuals with limited power or voice. Sustainable consumption issues surrounding TASFs have largely been covered in high-income contexts by academia and media particularly with a focus on bovine red meat. The evidence from this perspective argues for urgent and transformative measures to drastically reduce TASF consumption to influence livestock production patterns. However, in some populations—delineated age, livelihood, or region—the consumption of animal source foods (ASFs) more broadly still plays a crucial role in nutrition security.

Box 1Terrestrial animal source foods definition and scope.Terrestrial animal source foods (TASFs) include all food products coming from terrestrial animals (mammals, birds, and insects). The following food groups were applied in this review:
eggs and egg products;milk and dairy products;meat and meat products;foods from hunting and wildlife farming;insects and insect products.
Sub-groups were included based on different species, and the review primarily focused on unprocessed TASFs. Aquatic animal source foods were outside the scope of this review.

Despite concerted efforts on the part of the global community, it is unlikely we will meet the Sustainable Development Goal nutrition targets by 2030. Stunting prevalence among young children under five years was 22.3% and wasting was 6.8% in 2022, while overweight affected 5.6% of children and 39% of adults, and obesity, 4.3% of children and 15.8% of adults [1,2]. Globally, over half of preschool-aged children (56%; 95% uncertainty interval 48–64) and two-thirds of non-pregnant women of reproductive age worldwide (69%; uncertainty interval 59–78) have at least one of three micronutrient deficiencies (iron, zinc and vitamin A). The vast majority of these vulnerable populations live in south Asia, sub-Saharan Africa, or east Asia and the Pacific [3].

Many populations around the world are experiencing a double and even triple burden of malnutrition with simultaneously occurring undernutrition, overweight, and obesity, and micronutrient deficiencies and anemia. Cardiovascular diseases account for most deaths from diet-related non-communicable diseases (NCDs), or 17.9 million people annually, followed by cancers (9.3 million), and diabetes (2.0 million including kidney disease deaths caused by diabetes). A total of 77% of all NCD deaths are in low- and middle-income countries [4]. Around the world, there are four crucial metabolic risk factors for diet-related NCDs: overweight and obesity; elevated blood pressure; hyperglycemia; and hyperlipidemia. Nutrient- and energy-dense animal source foods, whether through lower or higher levels of intake, contribute significantly to most of these conditions.

This narrative review originated from an initiative of the Food and Agriculture Organization of the United Nations to produce “a comprehensive, science, and evidence-based global assessment of the contribution of livestock to food security, sustainable agrifood systems, nutrition, and healthy diets” [5]. In a three-paper series, we comprehensively cover the following: (1) the nutritional quality of TASFs by species and livestock system characteristics [6]; (2) the TASF effects on human health in the life course and relevant topics of allergies and microbiome (the present paper); and (3) the nutrition equity of TASFs including the enablers and barriers to adequate consumption among vulnerable populations. In the final paper, we overview food-based dietary guidelines [7] and explore the role of TASFs in different guidelines as further research.

### 1.2. TASFs in the Life Course and Contributions to Health

Humans, like other mammals, are heterotrophs and must acquire energy and nutrients from the environment. Biologically, we are classified as omnivores consuming plants and animals among other organisms, though social and environmental determinants can influence the shape of the diet. Dietary guidelines around the world, derived from evidence for links to human health, recommend healthy dietary patterns consisting of a diverse array of fruits, vegetables, nuts, and legumes, and TASFs and aquatic foods. Here, we cover evidence on the importance of TASFs in human biology. We briefly overview TASFs in hominin evolution to recognize the influence of the gatherer–hunter diet on our genome. Then, in greater detail, we present results from a narrative review for TASFs and health during vulnerable periods of the life course: pregnancy and lactation; childhood; and older adults.

#### 1.2.1. Animal Source Foods in Hominin Evolution

The evidence shows TASFs and animal source foods more broadly was present in hominin diets for over two million years and were driving factors of important physiological changes [8,9,10,11]. The two distinguishing elements of the gatherer–hunter–fisher diet relative to modern *Homo sapiens* were greater energy supplies from TASFs and diet diversity [9]. One estimate concludes approximately one-third of dietary energy intakes in gatherer–hunter–fisher diets came from meat consumption [12]. Consensus among several scholars is that diets rich in TASFs allowed for migration and adaptation to new environments [13,14,15].

At various points of hominin evolution, considerable anatomical and physiological changes occurred with evidence from suggesting ASFs in the diet or lack thereof played an important role. The first significant shift unfolded approximately two million years ago with the appearance of *Homo erectus*, marked by notable growth in height, body mass, and brain size, which were partly due to higher levels of ASFs in their diet [16,17,18,19,20]. Other physiological changes from other primates point to TASF nutrition (intake, absorption, and metabolism), including the lost capacity to absorb vitamin B12 derived from gut bacteria [21]; preferential absorption of heme iron versus non-heme iron [22,23]; greater dependency on dietary choline [21,24]; reduced capacity to produce taurine from amino acid precursors [25,26]; and lowered levels of alpha-linolenic acid conversion into eicosapentaenoic acid (EPA)/docosahexaenoic (DHA) [27].

A preponderance of evidence points to dramatic shifts away from ASFs to plant-based diets as our species moved to agricultural practices on a large scale during the Neolithic era. Skeletal evidence suggests that for many populations, the shift to agriculture was linked to poorer nutrition, reduced height, shorter lifespans, a higher burden of infectious diseases (due to increased population density), and a rise in dental caries, among other health issues [28]. All previously described evidence comes largely from the remains of adult hominins. A recent study uniquely examined the evidence for child dietary patterns from our ancestral past [29]. The evidence for child diets from gatherer–hunter–fisher (GHF) groups (*n* = 52) were compared to those from early agriculture (EA) groups (*n* = 43). TASFs together with aquatic animal foods were the most common food groups cited across subsistence modes, with higher mention frequencies in the GHF compared to EA. Climate zone was a key driver of food patterns with foods provisioned from local ecosystems.

In sum, evidence supports an important role for TASFs in dietary patterns and anatomical features during hominin evolution. This long evolutionary history of TASFs in diverse diets points to its importance in human biology. Social and environmental conditions today, however, perturbed the availability and access to ASFs around the world.

#### 1.2.2. TASFs in the Life Course

Nutritional requirements vary throughout the life course. Here, we use the phrase “life course” to capture both social and biological factors building from an original framework published by the Sub-Committee on Nutrition (ACC/SCN) [30] (Figure 1). A comprehensive review was conducted for all phases of the life course, but here we present findings from vulnerable periods with special nutrition considerations to support the growth, development, or the aging processes. The public discourse on TASFs has largely focused on adults, neglecting the vital phases of childhood, especially during preschool and school age and adolescence, and older adults.

## 2. Methods

Our review focused on the four phases of the human life course with special nutrition considerations: pregnant and lactating women; infants and young children during complementary feeding for both breastfed and non-breastfed children; preschool, school-age children and adolescents; and older adults. Using the Population Intervention/Exposure Comparator Outcome (PICO/PECO) framework, the criteria for inclusion were as follows:Population—generally healthy populations categorized by specific life course stages with special nutrient needs. Studies were eligible if the sample populations met the following criteria for life course phase: (1) pregnant and lactating women, including fetuses and breastfeeding children up to six months of age; (2) infants and young children (6–23 months); (3) children and adolescents (2–18 years); and (4) older adults (65 years or older). Some studies reviewed encompassed samples from multiple life course phases, and their findings were discussed in the relevant sections.Intervention/exposure—intake of TASFs and/or their products at various levels (e.g., high versus low intake; presence versus absence; one TASF type versus another TASF type). Preference was given to experimental trials to minimize risk of bias, though high-quality observational studies were included if potential confounding factors were accounted for in analyses.Comparator—control group (generally represented as the usual diet); lower levels of TASFs; higher levels of TASFs; plant-based foods or other dietary patterns.Outcome—health outcomes including nutrition were assessed, with variations based on life course stages, including anthropometric measurements, growth indicators, biomarkers of nutritional and health status (such as hemoglobin levels and nutrient biomarkers), measures of cognitive and neurological function and development, signs of food allergies and hyper-sensitivities, rates of infectious and chronic diseases, and all-cause and cause-specific mortality rates. Effect sizes with 95% confidence intervals were included when available. Studies dating from 2000 to the present were eligible; high priority was given to studies conducted in the past five years. The criteria for exclusion were the opposite of those for inclusion. We excluded studies examining immunocompromised individuals; populations requiring therapeutic diets; medical case reports; aquatic foods only; isolated TASF components; fortified TASF products; and TASF ingredients in processed foods.

All food products derived from mammals, birds, and insects were included in the literature search and were classified as the following: eggs and egg products; milk and dairy products; meat and meat products; food from hunting and wildlife farming; and insects and insect products (FAO 2023a).

The search and screening process included both peer-reviewed articles and the grey literature, utilizing a set of predefined search terms across various databases such as Medline, PubMed, EMBASE, different institutional websites, and Google Scholar (Appendix A). The retrieved articles, documents, and reports were organized by type into systematic reviews, other reviews, observational studies, intervention trials, and the grey literature (Figure 2). Two investigators screened the records identified and excluded those not eligible for inclusion and then screened the full-text articles, with inputs provided by an additional two investigators on the team. Consensus was reached by the team for studies ultimately included in the review. Since the initial review was carried out in 2020–2021, we updated the review with studies carried out in the previous three years. This review gave preference to high-quality systematic reviews of experimental studies or observational studies with low risk of bias. Context-specific studies were also highlighted when highly relevant and providing strong evidence.

Two additional topics were reviewed here as special considerations regarding TASFs and human health outcomes: (1) food sensitivities and allergies; and (2) TASFs and the human gut microbiome. While the evidence base is relatively smaller and less well representative of multiple populations compared to other health outcomes assessed, similar methods were followed in terms of the review of the literature.

## 3. Results

There were 155 studies included in this review. The findings from summarized systematic reviews are presented in Table 1, with positive outcomes represented by green arrows and negative health outcomes by red arrows. Table 1 includes the results of the review aggregated by type of TASFs and how they positively or negatively affect health within the vulnerable life course stages covered. In some cases, the quantity of TASFs consumed was linked to the health outcome, while other studies merely assessed whether or not TASFs were in the diet. The findings are discussed in greater detail in the sections below.

Generally, the evidence showed positive health outcomes across life course phases with special nutrient needs with some exceptions. We found a preponderance of evidence for milk and dairy products largely for improved health effects, with the exception of the finding for increased risks for large-for-gestational age among pregnant and lactating women [31] and prostate cancer associated with the consumption of low-fat milk and butter in older adults [32]. Red meat also showed positive outcomes across life course phases, except an increased risk for gestational diabetes among pregnant and lactating women [33]. The details for study design and effect sizes follow in the sub-sections by life course phase.

**Table 1 nutrients-16-03231-t001:** Summary of review findings for terrestrial animal source foods and health outcomes over the life course ^1^.

TASFs ^2^	Pregnancy & Lactation	Infants & Young Children 6–23 MO	Children 2–18 Y	Older Adults (65+ Y)
	Health Outcomes	Health Outcomes	Health Outcomes	Health Outcomes
**TASF, in general**	🡫 Maternal zinc deficiency [34]	🡩 LAZ [35,36,37,38] 🡩 WAZ [32,34,35] 🡩 WHZ [38] 🡩 Zinc and retinol status [38]		
**Dairy**	🡫 LBW, SGA/🡩 WAZ [36,37,38,39,40] † 🡩 Birth length/LAZ [36,37,39] † 🡩 Head circumference [38,41] † 🡩 LGA [31]	🡩 LAZ [37]	🡩 HAZ [42,43,44,45] 🡫 Obesity [44,45] 🡩 Bone mineral content and density [42,46] **Milk** 🡩 B12 concentrations [43] 🡩 MUAC and arm muscle area [43] 🡫 Overweight [43,44,47] 🡫 Obesity [43,44,47]	🡫 Sarcopenia/🡩 Skeletal muscle mass [48,49] 🡩 Cognitive outcomes [50] 🡫 Risk of dementia/Alzheimer’s disease [51,52,53] **Whole milk** 🡩 Risk of prostate cancer [32] **Low-fat milk** 🡩 Risk of prostate cancer [32] **Butter** 🡩 Risk of prostate cancer [32]
**Eggs**		🡩 LAZ [37]	🡩 Bone mineral content [53] †	
**Red meat**	🡫 Maternal risk of anemia [54] 🡩 Maternal risk for gestational diabetes [33]	🡩 LAZ [32,34]	🡩 Cognitive outcomes [55] † 🡩 B12 concentrations [56,57] † 🡩 MUAC [55] †	**Lean red meat** 🡩 Skeletal muscle mass [58]
**Insects and insect products**		**Caterpillar** 🡩 Hgb concentrations [59] †	**Honey** 🡩 LAZ [60] † 🡩 WAZ [60] † 🡩 WHZ [60] †	

† Indicates a one study citation. All other citations included are systematic reviews and/or meta-analyses. ^1^ A green upwards arrow (🡩) indicates a positive health finding for increased levels, while a green downwards arrow (🡫) indicates a positive health finding for decreased levels. A red upwards arrow (🡩) indicate a negative health finding for increased levels, while red downward arrow. Abbreviations: ^2^ Terrestrial animal source foods (TASFs); low birth weight (LBW); small for gestational age (SGA); weight-for-age z-score (WAZ); length-for-age z-score (LAZ); large for gestational age (LGA); weight-for-height z-score (WHZ); hemoglobin (Hgb); mid-upper arm circumference (MUAC).

### 3.1. Pregnant and Lactating Women, Including Fetuses and Breastfeeding Children up to Six Months of Age

Essential nutrients that are highly bioavailable in TASFs are needed in greater amounts during pregnancy and lactation to support fetal development, increased blood plasma volume, and milk production [61]. Malnutrition in this life course phase can result in serious consequences for both mother and offspring [62,63,64,65]. Iron and calcium requirements are very high during pregnancy and lactation, and deficiency may result in complications for the mother and fetus during gestation, on maternal bone resorption, and on offspring health, brain development, and cognition [65,66,67,68].

Intervention studies that examined TASF consumption during pregnancy and lactation have predominantly focused on milk and dairy products. A systematic review and meta-analysis evaluating the impact of milk and dairy intake during pregnancy on perinatal nutrition outcomes (such as anthropometry) [31] aggregated data from 14 studies involving a total of 111 184 pregnant (conducted in one low-income country, one lower middle-income country, one upper middle-income country, and eight high-income countries) and demonstrated that a higher intake of milk and dairy products, as compared to lower intake or none, was associated with increased birth weight (mean difference = 51.0 g, 95% CI: 24.7–77.3). The meta-analysis showed milk and dairy intakes increased infant length (mean difference = 0.33 cm, 95% CI: 0.03–0.64). Higher milk and dairy intake correlated with reduced risks of both small size for gestational age (odds ratio = 0.69, 95% CI: 0.56–0.84) and low birth weight (odds ratio = 0.63, 95% CI: 0.4–0.84), though increased the risk of large size for gestational age (odds ratio = 1.11, 95% CI: 1.02–1.21). No significant effect was observed on standard ultrasound measurements of fetal size.

Another systematic review investigated health (birth outcomes) and nutrition (breastmilk composition and anthropometry). This review drew from 17 studies: three intervention, three retrospective cohort, six prospective cohort, and two case-control studies, with six conducted in low- and middle-income countries and eleven in high-income countries. The findings of this review were consistent with the previous review, showing a positive association between milk intake during pregnancy and infant birth anthropometry (weight and length) [39]. Evidence was inconclusive regarding the effects of milk and dairy intakes during pregnancy on birth outcomes including preterm birth and spontaneous abortion or on breastmilk composition during lactation.

Prospective cohort studies, the Generation R Study and its continuation, Generation R Next, in the Netherlands tracked participants from fetal development to young adulthood [69]. An analysis within these studies evaluated the relationship between fetal growth, neonatal complications, and milk consumption in the first trimester (*n* = 3405 mothers) [40]. The results of a semi-quantitative food-frequency questionnaire indicated consumption of 450 mL or more of milk each day, compared to lesser amounts (150 mL or 0 mL) of milk consumption each day, was associated with healthy and greater fetal weight gain in the third trimester and an 88 g increased healthy birth weight (95% CI: 39–135 g). Infants of mothers who consumed three or more glasses of milk daily had head circumferences that were 2.3 cm larger compared to those whose mothers drank one or zero glasses of milk. There was no significant association for infant length. Another prospective cohort study in Denmark assessed how milk consumption during pregnancy affected the offspring’s nutritional status (anthropometry) at birth, as well as health outcomes (IGF-1 levels, insulin levels) over a follow-up period of approximately 20 years (n = 685) [41]. The study found that at 30 weeks gestational age, mothers drinking 150 mL/day or more compared to those drinking less than 150 mL/day was associated with birth outcomes for the offspring showing a greater length-for-age Z score (0.34, 95% CI: 0.04–0.64) and weight-for-age z-score (0.32, 95% CI: 0.06–0.58) within a healthy range. A 20-year follow-up of the children born to mothers with high milk intake revealed that these children had higher height z-scores and higher levels of IGF-1 and insulin when compared to those born to mothers with lower milk intake; however, these differences were not statistically significant. An observational study [55] among pregnant Portuguese women (*n* = 98) identified an association between total dairy consumption (including milk, yogurt and cheese) in the first trimester and both placental weight (beta = 0.333, *p* = 0.012) and head circumference (beta = 0.002, *p* = 0.014).

Additional observational studies were included to enhance the population representativeness of this review. One prospective cohort study [70] focused on pregnant women residing in an urban area of South India and investigated how milk and dairy product consumption affected birth nutrition outcomes (anthropometry) using a food-frequency questionnaire (n = 2036). The study found that mothers who consumed total milk (including whole and semi-skimmed milk) and dairy products and percentage of protein from milk in the highest quintile (557.7–731.0 g, 30.3–36.9%) had neonates with a greater birth weight than those mothers who consumed in the least quintile (61.4–144.6 g, 12.8–15.1%) (beta = 86.8, 95% CI: 29.1–144.6; beta = 63.1, 95% CI: 10.8–115.5; *p* < 0.001 for both).

Beyond milk and dairy products, there is evidence suggesting that the consumption of other TASFs can impact health outcomes during pregnancy and lactation. A systematic review with meta-analysis [54] found that pregnant women consuming meat ≤ 1 time per week had over twice the risk of anemia (OR 2.02; 95% CI 1.55, 2.50). Similarly, a review of four studies from Ethiopia [34] indicated that zinc deficiency was linked to low intakes of animal-source food (OR 2.57; 95% CI 1.80, 3.66). The research has also established an association between red meat consumption and increased risk of gestational diabetes mellitus from high-income countries [33].

In sum, the findings consistently demonstrate that milk consumption during pregnancy increases birth length, weight, and head circumference, with some evidence suggesting that initiating the intervention or milk intakes in the first trimester may increase the strength of these effects. Recent evidence suggests the potential for protective association between meat and anemia prevention. Generally, there is a lack of evidence on other TASFs beyond milk and dairy products and from LMICs.

### 3.2. Infants and Young Children (6–23 Months)

The complementary feeding period refers to the time between 6 and 23 months when an infant or young child continues breastfeeding but needs additional foods to fulfill their daily nutritional needs [2]. During this period, TASFs may play a critical role as a source of bioavailable nutrients that can be absorbed more efficiently, especially considering the limited gastric capacity and the need to support neurological development and rapid growth [71,72,73,74]. At this stage of the life course, the child’s immunity is shifting from relying on the passive immunity provided by the mother during pregnancy and lactation to developing their own independent, maturing system. Zinc, which is abundant in TASFs, is a key nutrient essential for the development of this adaptive immunity [75]. Grounded in a strong evidence base, the new WHO Guidelines for Complementary Feeding now recommend animal source foods be consumed daily in the context of a diverse healthy diet [76].

Several systematic reviews have explored the impact of TASFs on nutrition outcomes in young children, particularly focusing on underweight and stunting. One review, which included six trials (*n* = 3036 children) from the United States, China, Pakistan, Zambia, the Democratic Republic of the Congo, Guatemala, and Ecuador [35], assessed the effects of TASFs on growth and development. The findings showed that in three studies, length-for-age z-scores or height-for-age z-scores were significantly increased in the intervention groups compared to the controls. Another three studies found TASF consumption among young children significantly increased the weight-for-age z-scores. Furthermore, the review included findings from a study on diets incorporating meat and dairy, which showed a significant increase in length-for-age z-score with meat consumption but not with dairy consumption, and found no significant changes in weight-for-age z-score [35].

Another review [37] of longitudinal panel data found strong relationships between any TASF consumption and higher length-for-age z-score in children aged 6–23 months in Nepal and Bangladesh. The higher the number of ASFs consumed, the stronger the association. When considering different food groups, dairy showed the most significant correlation with growth in Bangladesh and Nepal. Meat intake in Nepal, fish consumption in Uganda, and egg consumption in Bangladesh also contributed to growth.

A systematic review [36] investigating the impact of ASFs on stunted growth among children aged 6 to 60 months in LMICs included 21 studies, comprising one randomized-controlled trial and one cross-sectional study that revealed significant decreases in stunting. Secondary outcomes of anemia, height/weight, and head circumference did not show significant changes.

The recent research has investigated the effects of complementary feeding of meat on infant micronutrient levels. A systematic review examining complementary feeding practices in high-resource settings found that meat consumption (from a variety of types including beef, chicken, pork, lamb, turkey, or cod) lowered the risk of iron deficiency in breastfeeding infants who had low iron intake or were at risk of insufficient iron stores during the first year. However, this effect was less pronounced in infants who already had adequate iron stores. There was limited evidence regarding the impact of meat consumption on zinc status in infants during complementary feeding [77]. Livestock keeping significantly contributed to increased TASF consumption according to another systematic review including 176 studies in low- and middle-income countries [38]. This heightened consumption was associated with improved height-for-age z-score, weight-for-height z-score, weight-for-age z-scores, and zinc and retinol status in children under 5 years of age. There were conflicting associations between anemia and TASF consumption.

Few studies have investigated the consumption of insects and their products in young children. In the Democratic Republic of the Congo, a study tested the use of caterpillar cereal as a complementary food and found that it positively affected hemoglobin concentrations and decreased the prevalence of anemia among young children [59].

In sum, high-quality evidence suggests that TASF consumption offers some benefits to infants and young children, including increases in height/length and weight. However, consideration for the overall dietary pattern and the specific contexts is needed when assessing and generalizing these relationships.

### 3.3. Children and Adolescents (2–18 Years)

Nutrition continues to be an important factor underlying health throughout childhood and adolescence. During this phase, neurogenesis and synapse formation occur at high rates followed by the significant pruning of excess neural connections and synapses during adolescence [78]. Brain development during childhood necessitates diets rich in energy and micronutrients, accounting for 50% of the body’s total basal requirements [79]. Similar to infancy and early childhood, TASFs can provide critical nutrients in bioavailable forms (e.g., iron, zinc, DHA, choline, vitamin B12) to meet the needs of brain development in middle childhood and adolescence. Infection can disrupt (via inflammation) nutrient absorption and utilization, with further implications for such outcomes as neurodevelopment similar to infants and young children [80].

Growth patterns in prepubescent boys and girls generally follow similar trajectories [81]. Around age nine, girls start their pubescent growth spurt, while boys typically begin this phase approximately two years later. During this period, differences in adiposity tissue and fat deposits related to biological sex become apparent [81]. Adolescence is also a critical time for achieving peak bone mass, requiring adequate intake and stores of nutrients including protein, zinc, vitamin D, calcium, phosphorous, magnesium, among others [82]. Evidence clearly supports the need for TASF intake as part of healthy diets to promote bone health during these stages of the life course [83]. With menarche, girls’ iron requirements increase [81,84]. TASFs provide the highly bioavailable nutrients needed for rapid growth and the processes involved in sexual maturation during adolescence in both boys and girls [85,86,87].

Impacts on growth, bone health, and cognition have been assessed in preschoolers, school-age children, and adolescents in relation to TASF consumption. Milk and dairy products among children aged 2–18 years for physical growth outcomes was assessed in a systematic review and meta-analysis [42]. Twelve studies from the United States, Europe, China, India, Indonesia, Kenya, and Vietnam showed a daily milk supplement of 245 mL resulted in a 0.4 cm annual increase in height compared to children who did not receive this supplement. Additionally, stunted children experienced greater height benefits from milk consumption than their peers and milk had a more pronounced effect on child height compared to other dairy products. Among adolescents, dairy consumption yielded the most significant height effects.

The alarming rise in the double and even triple burden of malnutrition necessitates reviewing the effects of TASFs on the risk of overweight and obesity in children and adolescents. Based on cross-sectional studies, the total dairy consumption in children aged 2–21 years is associated with a reduced risk of obesity (OR 0.66; 95% CI 0.48, 0.91) but not overweight [47]. Similarly, an additional meta-analysis of cross-sectional studies found that milk consumption was associated with lower odds of obesity in children 4–18 years of age (OR: 0.87, 95% CI: 0.80–0.95) [45]. In a meta-analysis of 14 studies (11 cross-sectional and 3 cohort), regular consumption of whole milk (compared to reduced fat milk) was associated with a lower risk of overweight and obesity among children aged 1–18 years (OR 0.61; 95% CI 0.52, 0.72) [49]. Milk and dairy product consumption were also found to be associated with a reduced risk of overweight and obesity in other systematic reviews with effect sizes ranging from 0.54 to 0.87 [43,45,47].

A systematic review explored the risks of overweight and obesity over the life course linked to child milk and dairy consumption in children [46]. This review included 10 cohort studies (n = 46,011 children and adolescents) with an average follow-up period of three years, conducted in the United States, the United Kingdom, Sweden, and China. The findings revealed that children with the highest dairy intake had a lower risk of overweight/obesity compared to those with the lowest intake (pooled odds ratio = 0.62; 95% CI: 0.49–0.80). Adjusted regression modeling analysis indicated that each additional daily serving of dairy was associated with a 13% reduction in the risk of overweight/obesity compared to lower dairy consumption (odds ratio = 0.87; 95% CI: 0.74–0.98).

With regards to insect products, in a small study conducted in Indonesia, young children aged 24 to 59 months (*n* = 60) were randomly assigned to two groups: one group received 45 g of honey daily for two months, while the other acted as a control [60]. There was some evidence for a positive effect on anthropometry in the intervention group compared to the control group, with differences evident for height-for-age z-score (MD 0.424, *p* ≤ 0.05), weight-for-age z-score (MD 0.186, *p* ≤ 0.05), and weight-for-height z-score (MD 0.118, *p* ≤ 0.05), that may justify replication in a larger trial.

The research in school-age children and teenagers has explored the impact of dairy and egg consumption on bone health. Eight studies were identified in a review of randomized controlled trials spanning across New Zealand, Germany, China, the United States, Finland, and the United Kingdom. Dairy consumption showed positive effects on bone mineral content and density [48]. Another meta-analysis of randomized controlled trials [44] found that general dairy supplementation increased bone mineral content (25.37 g, 95% CI: 7.50, 43.25 g), mineral density (0.016 g/cm^2^, 95% CI: 0.006, 0.025 g/cm^2^), and height (0.21 cm, 95% CI: 0.09, 0.34 cm) of children and adolescents aged 3 to 18 years. In an observational study conducted in the United States, investigators assessed the relationship between egg intake and bone health in 13-year-olds (*n* = 294), finding a positive association specifically for radium cortical bone mineral content and biomarkers of bone metabolism such as osteocalcin [88].

Despite the strong evidence that the highly active brain development during childhood and adolescence could be responsive to TASF consumption, there has been limited research on this connection. A systematic review sought to explore how beef and beef product consumption affects cognitive outcomes in children and young adults [89]. The trials in this review compared beef with other TASFs (such as a glass of milk), or with plant-based foods. Among these, only one intervention, that compared beef consumption to a usual-diet control group, demonstrated positive effects on cognitive outcomes [43].

The findings from a widely cited cluster randomized-controlled trial conducted in Kenya point toward the benefits of TASF nutrition in school-age children [56,90,91,92]. This trial assessed the health and development outcomes of school-age children (*n* = 911) across four groups: one receiving meat with githeri stew (maize, beans and greens); one receiving milk with githeri stew; one receiving githeri stew alone; and a control group. The results showed that the children in the meat group experienced a greater increase in Raven’s Progressive Matrices scores (which measure fluid intelligence, perceptual awareness, problem-solving, abstract reasoning, perceptual awareness, and reasoning) compared to all the other groups. Among those in the milk group, only younger children and those who were stunted at baseline displayed a greater rate of height gain than their peers in the same group [43]. The plasma concentrations of vitamin B12 were elevated in both the meat and milk groups [56,57]. Furthermore, mid-upper arm circumference and arm muscle area significantly increased in the meat and milk groups compared to the other two groups [93]. Variations in morbidity rates and other outcomes were also observed [94].

In sum, preschoolers, school-age children, and adolescents experience critical growth and developmental changes, including those related to reproduction, endocrine function, and neurodevelopment. These processes demand diets rich in energy- and nutrient-dense foods. The evidence indicates that milk and dairy products can increase height, improve bone density and mineral content, and reduce the risk of overweight and obesity. Milk and meat consumption were linked with positive cognitive and health outcomes in a trial in Kenya.

### 3.4. Older Adults (Age 65 and Older)

Older adults, particularly those in low-resource settings, are often overlooked in public health nutrition initiatives. However, the aging process involves several physiological changes that may impact dietary requirements [95]. As cells enter senescence or growth arrest, protein expression is altered, and oxidative stress increases, resulting from the accumulation of iron and other inflammatory factors at the cellular level. Inflammatory responses, alterations in blood flow, and neurotransmitter metabolism can affect cognition and memory in aging. The evidence points to the potential for TASFs to mitigate some of these challenges. For example, choline found in high concentrations in TASFs has been shown to offer neuroprotective benefits including memory, making it particularly relevant to healthy aging [96].

Sarcopenia, which refers to the loss of lean body mass or muscle during aging, is influenced by changes in protein metabolism over time in which protein synthesis fails to compensate for protein catabolism [97,98]. It has been hypothesized that meat, rich in bioactive compounds like creatine and carnitine, may help address these issues [99]. A narrative review encompassing 11 articles (five reviews, five clinical trials, and one cluster RCT) showed older adults consuming 113 g of meat five times a week would be the optimal dietary intake to mitigate sarcopenia [100]. Similarly, TASFs may prevent osteoporosis and safeguard bone health, a condition that arises when the body is unable to regenerate new bone tissue effectively. At the age of 65, the dietary requirements for certain minerals, particularly magnesium and calcium, increase. Together with vitamin D, these minerals are vital for the prevention of bone loss and related frailty fractures in older adults [101]. Moreover, TASFs rich in DHA, such as eggs, may play a crucial role in preventing macular degeneration and brain disorders, improving memory, and providing broader neuroprotection [102].

The impact of milk and dairy consumption on nutrition, health, and cognitive outcomes in older adults has been explored, particularly in relation to sarcopenia, frailty, and cognitive decline (e.g., Alzheimer’s disease). A narrative review of six cohort studies conducted in Australia, China, France, Japan, and the United States of America found that a high dairy intake was linked to a reduction in frailty and sarcopenia by improving skeletal muscle mass [50]. The review also highlighted two prospective-cohort studies [51,58] from Japan that showed beneficial effects on cognition, though the overall cognitive outcomes across all studies were mixed. Another literature review focused on the potential of milk and dairy consumption to prevent Alzheimer’s disease and other forms of dementia [51]. Once more, the two Japanese prospective cohort studies provided evidence of positive effects—one study indicated that frequent milk consumption (almost daily compared to less than four times per week) lowered the odds of vascular dementia [52], while the other study found that increased milk and dairy intake reduced the risk of Alzheimer’s disease [53]. A positive association has been observed between the risk of prostate cancer and total milk (including both whole and semi-skimmed milk) intake (RR: 1.07, 95% CI: 1.00, 1.14) in a meta-analysis of 33 studies [32]; however, an inverse association was seen between the risk of prostate cancer and whole milk intake (RR: 0.93, 95% CI: 0.87, 0.99) while skim/low-fat milk intake increased the risk (RR: 1.10, 95% CI: 0.96, 1.26). Butter also had a positive association with prostate cancer risk (RR: 1.08, 95% CI: 1.03, 1.12), and cheese, yogurt, cream, and ice cream had no significant association with prostate cancer risk.

A systematic review [58], including nine intervention and nineteen observational trials examining whole foods for effects on sarcopenia and muscle health, reported robust and consistent findings for the beneficial effects of consumption of dairy-food ≥2 servings and lean red meat on muscle mass or lean tissue mass. Higher consumption of processed meats increased the risk of poor physical functioning over time. The evidence for eggs and fish was inconclusive. To our knowledge, no studies have compared dietary patterns with different TASFs among older adults.

In sum, the epidemiological evidence of TASF consumption and their effect on the health of older adults comes primarily from high-income countries. Consistent strong findings indicate positive effects on muscle health associated with lean red meat consumption. Additional evidence indicates that milk, dairy products, unprocessed meat, and eggs have potential in preventing frailty, fractures, sarcopenia (muscle loss), dementia, and Alzheimer’s disease.

## 4. Special Considerations

Two topics were considered in this review with important relevance to TASFs and human health: food sensitivities and allergies; and the human gut microbiome. These special considerations were handled as a narrative review, distinct from the systematic review methods applied for the life course section.

### 4.1. Food Sensitivities and Allergies

The CODEX Alimentarius Expert Committee on Food Allergens [103] indicates eggs and milk represent a risk in terms of food allergies. With regards to milk, sensitivity is due either to an adverse reaction to milk proteins, particularly cow’s milk proteins or the non-digestion of lactose, the key carbohydrate found in mammal’s milk. The latter, with the exception of congenital lactase deficiency which is a rare genetic disorder [104], is not a disease, but a normal variant in human metabolism. Lactase deficiency is the inability to digest large amounts of lactose; lactase, the enzyme that allows the digestion and absorption of lactose, is present in higher concentration during infancy [105]. Lactase deficiency, the main cause of lactose malabsorption, can lead to lactose intolerance, though not in all individuals.

The prevalence of lactose malabsorption globally is estimated to be 68% among individuals older than 10 years of age. However, there are significant differences across regions and to greater extent among countries [e.g., the Middle East (70%); Western, Southern, and Northern Europe (28%)]. Differences in assessment methods, thresholds, or non-representativeness of the data might explain these important variations. It is interesting to note that native populations showed different prevalences, possibly due to a selection process that allowed these populations for whom cattle are important to better digest dairy products [106]. With regards to cow’s milk allergy (CMA), the prevalence in high-income countries at 1 year of age ranges between 0.5 and 3% [107], between 0.6 and 2.5% among preschoolers, 0.3% for older children and teens, and 0.5% for adults [108].

Egg hypersensitivity, one of the most common food allergies, is due to an adverse reaction to the protein ovomucoid and affects 1.7% of the population [109]. Both individuals with cow’s milk allergy and those with egg hypersensitivity tend to outgrow the allergy between 2 and 5 years of age [110] and before they reach school age or during their late teens [109]. For this reason, the 2023 WHO Guidelines on complementary feeding between 6 and 23 months, have not included any recommendation to avoid intake of animal milk (in the absence of breastfeeding) and eggs in the 6–12 month period and later.

In sum, while eggs and milk represent a potential risk for food allergies with serious health implications, the prevalences are low and vary across contexts. More evidence with standardized assessment methods and greater representation of different populations would improve our understanding of TASF sensitivities and allergies.

### 4.2. TASFs and Human Gut Microbiome

The microbiome, specifically in the gut, is integral to human health through the life course. The quantity and source of TASFs consumed influence the microbiome’s composition and microbial metabolites produced, with potential consequences for health [111,112]. The human gut microbiome differs between vegan/vegetarian and omnivores populations, the latter having a lower diversity, and less of the beneficial *Faecalibacterium prausnitzii* [113,114,115]. Both short and long-term interventions in humans with a diet high in TASFs increased the abundance of pro-inflammatory species [116,117,118]. Each TASF seems to influence the gut microbiome differentially [119,120]. In rats, the beneficial *Lactobacillus* genus was present at a higher level with the consumption of white meat compared to red meat [119]. In mice and in humans, beef consumption decreased the relative abundance of *Bacteroidetes* and increased *Firmicutes* and *Proteobacteria* (among which are some potential pathogenic bacteria) [119,121,122]. Fried meat and processed meat are associated with negative changes in the microbiome [117,123,124]. The intake of dairy products generally led to a positive influence on the gut microbiome, enhancing bacterial diversity and boosting the presence of beneficial genera such as *Lactobacillus* and *Bifidobacterium* [116,125,126,127,128]. In human studies, a diet rich in fermented food, including both dairy and plant products, has been demonstrated to lower inflammation by modulating the microbiome [129,130].

The differences observed between varying types of TASFs on the gut microbiome may be the result of different amino acid compositions as well as in fat and fiber contents [120]. The bacterial metabolism of TASF dietary proteins produces several types of metabolites, some beneficial such as short-chain fatty acids and other metabolites associated with either positive or negative health outcomes (i.e., trimethylamine N-oxide (TMAO) and toxic nitrogenous and sulfur metabolites) [131,132]. Carcinogenesis might be influenced by red and processed meat consumption, with some effects potentially mediated by metabolites from the microbiome [133,134,135]. A high circulating TMAO concentration correlates with higher risk of cardiovascular diseases [136], diabetes [137], and obesity [138] in adult populations. In humans, diets rich in animal proteins lead to the elevated production of TMAO [135,139,140,141]. Different TASFs affect plasma TMAO differently: red meat (carnitine) is correlated with higher TMAO levels [142,143], but not white meat, eggs, or dairy products [144]. In animal studies, diet-induced TMAO production by the microbiome exacerbated hepatic steatosis [145], atherosclerotic development [146], and thrombosis [147]. Importantly, the overall quality of the diet can either mitigate or exacerbate the impact of TASFs on the microbiome [132,148,149,150]. Further insights into mechanisms underlying the impact of specific ASFs on the gut microbiome are needed in humans.

## 5. Discussion and Synthetic Conclusions

Here, we reviewed the evidence base for the dietary intake of TASFs on human health, applying a life course framework to acknowledge varying nutritional needs arising from such processes as growth, development, reproduction, and aging. Too often, the public discourse on TASFs and health defaults to adult populations, failing to consider the special nutrient needs in other life course phases. This analysis, focused largely on the biology and epidemiology of TASFs and health, is one of three papers emerging from the assessment conducted by the Food and Agriculture Organization of the United Nations to comprehensively examine the evidence base for TASF. This paper was the second of the three-paper series described previously: (1) nutritional quality of TASFs by species and livestock system characteristics [6]; (2) TASF effects on human health in the life course and relevant topics of allergies and microbiome; and (3) nutrition equity of TASFs including the enablers and barriers to adequate consumption among vulnerable populations and TASF role in dietary guidelines as further research.

The recent reviews on ASFs have either focused on certain phases of the life course such as infants and young children, or distinct TASFs such as milk, meat, or eggs. We comprehensively covered all TASFs and uniquely included life course phases with special nutrient needs. Populations living through these phases require nutrient-dense foods to undergird the biological processes and maintain health. The high bioavailability of nutrients in TASFs, as covered in the first paper of this series, allows for more efficient absorption and metabolism necessary during rapid growth and development. Across all life course phases, we found that milk and dairy products consistently showed positive health impacts. Increasing access to these foods for vulnerable groups, as addressed in the third paper of the series, could lead to important public health impacts globally. Similarly, contextual and environmental factors can mitigate the human health effects of other TASF.

To supplement our systematic review of the literature for life course phases, we included brief narratives on relevant topics. TASFs in human evolution played a critical role in driving brain growth and development and other physiological and anatomic features, ultimately influencing our genomes for health outcomes over time [8]. Cow milk and eggs represent a risk in terms of food sensitivities and allergies, though the prevalence in high-income countries (where there is most of the evidence) is low, ranging from 0.5 to 3% for cow’s milk and 1.7% for egg sensitivities, and individuals often outgrow these allergies. TASFs have been found to affect the human microbiome composition and microbial metabolites produced. Insights gleaned from this evidence suggest mechanistic pathways through which TASFs can influence health outcomes, both positive and negative, via the gut microbiome [151]. Some evidence suggests a reduction in beneficial gut organisms and increase in pro-inflammatory species with the higher consumption of TASFs, though varying by type and quantity consumed. Dairy products showed positive effects on the gut microbiome by increasing bacterial diversity and the abundance of beneficial genera, and fermented products more broadly appear to reduce inflammation modulated by the microbiome.

This review had some limitations. First, we covered terrestrial animal source foods originating from assessment of livestock-derived foods and human health, to the exclusion of aquatic ASFs. Aquatic foods similarly provide important nutrients in bioavailable forms and can protect nutrition during vulnerable periods of the life course [152]. Another limitation in our review is the potential gap in the assessment by excluding non-English language papers. There were also limitations in the evidence base that speak to future research needs. The preponderance of the evidence assessed examined milk and dairy products. Beef and eggs, and to a lesser extent white meat, follow in terms of availability of evidence, with few trials on insects and their products. This review did not find any specific study on pig, poultry meat, meat from wild animals, and meat from other livestock species such as goats, sheep, and rabbit. In terms of vulnerable life course phases, we found the evidence base concentrates more heavily on women during pregnancy and lactation and infant and young children, followed by preschool and school-age children and adolescents and significantly less on older adults, especially in LMICs.

Ultimately, evidence supports a role for TASFs in healthy, diverse diets across the life course, particularly the periods when nutrient needs increase or change. Moving forward, there is an imperative to fill gaps in the evidence base across multiple contexts and life course phases and consider proportions and quantities to inform dietary patterns and guidelines globally.

## Figures and Tables

**Figure 1 nutrients-16-03231-f001:**
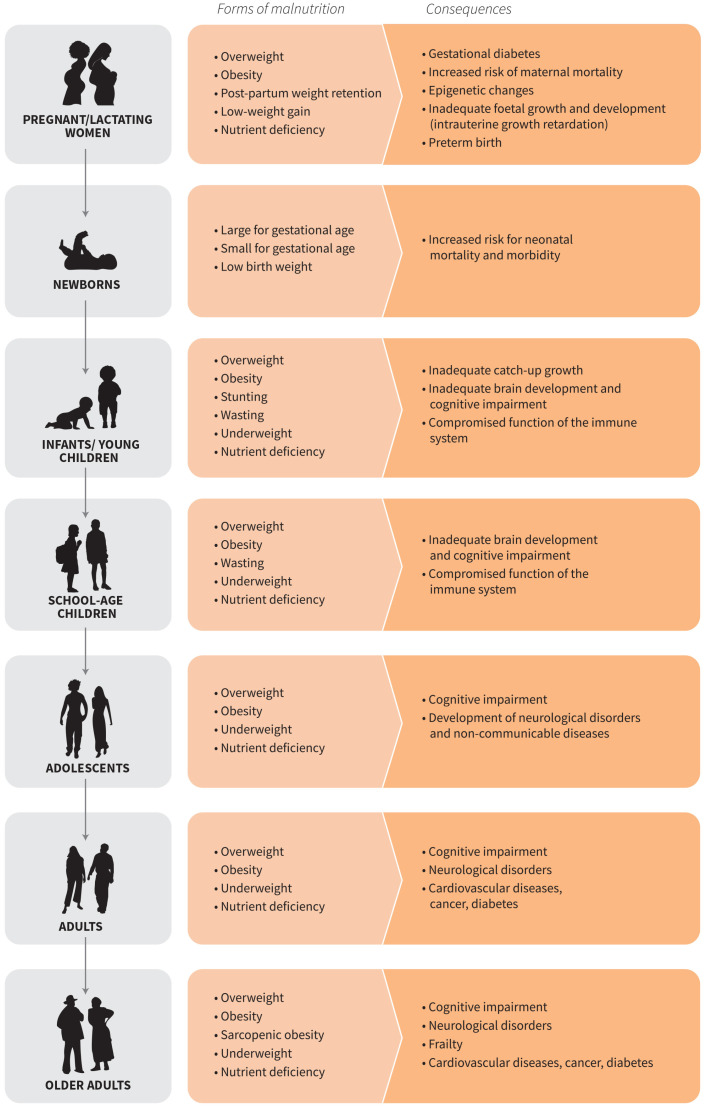
Forms and consequences of malnutrition over the life course (FAO 2023a).

**Figure 2 nutrients-16-03231-f002:**
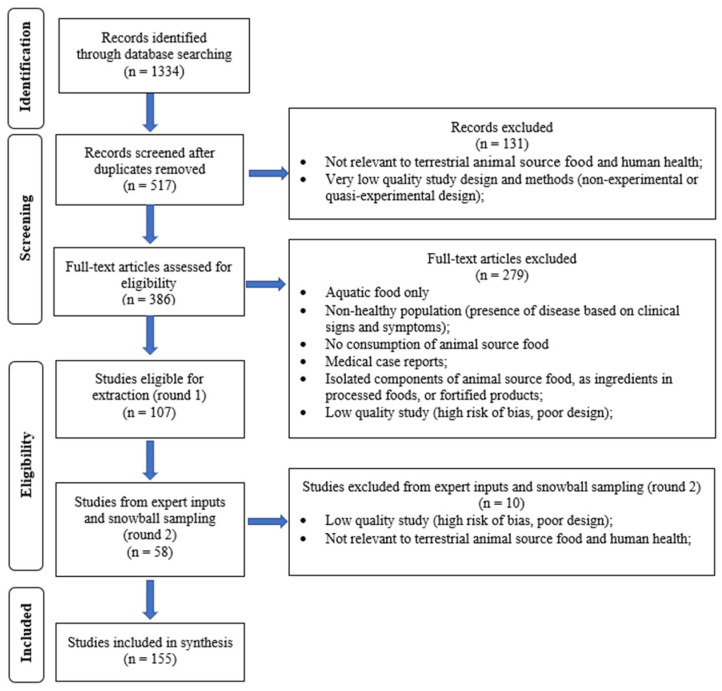
Terrestrial animal source foods and health outcomes—study selection.

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
