# Peer review of "Terrestrial Animal Source Foods and Health Outcomes for Those with Special Nutrient Needs in the Life Course"

_nutrients, 2024, doi:10.3390/nu16193231_

Round 1

Reviewer 1 Report

Comments and Suggestions for Authors

Line 28

Is there a clear definition and scope of TASF?

Line 107-112

Is there any evidence or research comparing anatomical and physiological changes between different dietary patterns in adults or older during the hominin evolution other than the child's diet?

Line 176

A detailed description of the arrows in Table 1 is required. Moreover, the format of the references listed in Table 1 does not meet the requirement.

Line 253-257

Is there any evidence that excessive consumption of milk or dairy products during pregnancy leads to fetal overweight or excessive head circumference?

Line 293 "Feeding of meat on infant"

What kinds of meat are mainly included in this part of the investigation?

Line 333-335

How does this dietary pattern affect the rapid growth and sexual maturation process of male development?

Line 414

Does any research exist that contrasts various dietary patterns with TASF, focusing on older adults' health?

Line 476

Conclusive statements are needed to illustrate the importance of food sensitivities and allergies concerning TASF in body health.

Comments on the Quality of English Language

none

Author Response

Manuscript ID: nutrients-3161525

Type of manuscript: Review

Title: Terrestrial animal source foods and health outcomes for those with special nutrient needs in the life course 

REVIEWER 1

We very much appreciate the thoughtful comments of Reviewer 1 and have worked to address all.  Kindly find our responses below.

COMMENT 1

Line 28

Is there a clear definition and scope of TASF?

RESPONSE 1

We have added Box 1 now to more clearly define TASF and scope of the review. 

COMMENT 2

Line 107-112

Is there any evidence or research comparing anatomical and physiological changes between different dietary patterns in adults or older during the hominin evolution other than the child's diet?

RESPONSE 2

Thank you for this observation.  The majority of evidence supporting ASF in hominin diets actually does come from adult populations leading up to the statement about child evidence. We added a sentence to more clearly specify this. Unfortunately, the evidence does not specify if the remains came from older adults, so we left it as a general category of adults.

COMMENT 3

Line 176

A detailed description of the arrows in Table 1 is required. Moreover, the format of the references listed in Table 1 does not meet the requirement.

RESPONSE 3

In response to this comment, we have added a footnote to Table 1 to provide a detailed description of the green and red arrows. The references listed in Table 1 have been formatted to ACS style to fit journal specifications.

COMMENT 4

Line 253-257

Is there any evidence that excessive consumption of milk or dairy products during pregnancy leads to fetal overweight or excessive head circumference?

RESPONSE 4

To our knowledge, these systematic reviews of studies did not show that over consumption of milk or dairy products lead to fetal overweight or excessive head circumference.  We returned to the reviews to check and found no evidence of this.  Thank you for raising your concerns in any case.

COMMENT 5

Line 293 "Feeding of meat on infant"

What kinds of meat are mainly included in this part of the investigation?

RESPONSE 5

The narrative presents finding from a systematic review, so we returned to the original studies and included the meats as specified – beef, chicken, turkey, lamb, pork, or cod – now added to the manuscript.

COMMENT 6

Line 333-335

How does this dietary pattern affect the rapid growth and sexual maturation process of male development?

RESPONSE 6

Thank for this question.  In fact, TASF are important for both boys and girls to support rapid growth and sexual maturation.  We added this phrase and included 3 more references to this sentence.

COMMENT 7

Line 414

Does any research exist that contrasts various dietary patterns with TASF, focusing on older adults' health?

RESPONSE 7

This would be intriguing to explore, but to our knowledge, no studies have compared dietary patterns with different TASF among older adults. We added mention of this in the manuscript.

COMMENT 8

Line 476

Conclusive statements are needed to illustrate the importance of food sensitivities and allergies concerning TASF in body health.

RESPONSE 8

Thank you for this comment.  We have added concluding statements to this section.

Reviewer 2 Report

Comments and Suggestions for Authors

Comments:

This important paper provides a literature review on terrestrial animal-source foods (TASF) and human health by life stage. Although the authors presented interesting research literature, they did not clearly show the key methodological aspects of a systematic review. Although they mentioned that they followed the recommendations of the PRISMA statement for systematic review (references 29 and 30), some descriptions need to be better addressed in their paper. 

Below, I present some key issues that have not been mentioned in the paper:

 1 - Abstract:

There was no mention of the number of included materials, the reporting of the summary estimate, the confidence/credibility of the meta-analysis, if done, or the limitations of the evidence.

2- Introduction.

In this section, the authors must present the rationale for the review in the context of terrestrial animal source foods (TASF) and human health by life stage, separated from the methods. 

3 - Methods

Methodological aspects need to be expanded and better described. For example, the timeframe and details of the eligibility criteria - such as the language of the material, assessment of the risk of bias of the study, effect measures, expert sampling procedures, instruments for summarising - the role of each participant and procedures included - were not described.

The authors mentioned that they developed a systematic review. However, in line 78, they mentioned their results: "Then in more detail, we present results from a scoping review for TASF and health during vulnerable periods of the life course: pregnancy and lactation; childhood; and older adults", which is a confusing statement.

The methods section should mention comments that are part of the special consideration (lines 471-475).

4 - results and conclusion

It must be revised according to references 29 and 30, with the study limitation mentioned before the conclusion.

5 - Supplementary material should include the registration and protocol and the availability of the data, code and other material.

Author Response

Manuscript ID: nutrients-3161525

Type of manuscript: Review

Title: Terrestrial animal source foods and health outcomes for those with special nutrient needs in the life course 

We very much appreciate the comments provided by Reviewer largely focused on upholding the rigor and standards of a systematic review and meta-analyses. While we worked hard to comprehensively and systematically assess the evidence-base, we did not conduct any meta-analyses and ultimately produced a narrative review. Nonetheless, we address as many comments as relevant below.   

This important paper provides a literature review on terrestrial animal-source foods (TASF) and human health by life stage. Although the authors presented interesting research literature, they did not clearly show the key methodological aspects of a systematic review. Although they mentioned that they followed the recommendations of the PRISMA statement for systematic review (references 29 and 30), some descriptions need to be better addressed in their paper. 

Below, I present some key issues that have not been mentioned in the paper:

COMMENT 1

 1 - Abstract:

There was no mention of the number of included materials, the reporting of the summary estimate, the confidence/credibility of the meta-analysis, if done, or the limitations of the evidence.

RESPONSE 1

Kindly note the general comment above; we did not carry out any meta-analyses.  We appreciate the comment and have revised the abstract to provide more detail on the limitations of the evidence.

COMMENT 2

2- Introduction.

In this section, the authors must present the rationale for the review in the context of terrestrial animal source foods (TASF) and human health by life stage, separated from the methods. 

RESPONSE 2

We agree that the introduction of the manuscript with the rationale for assessing TASF by life phase should be distinct from the Methods section.  The manuscript has been reordered with the rationale included in the first section preceding the Methods.  Thank you for these comments.

COMMENT 3

3 - Methods

Methodological aspects need to be expanded and better described. For example, the timeframe and details of the eligibility criteria - such as the language of the material, assessment of the risk of bias of the study, effect measures, expert sampling procedures, instruments for summarising - the role of each participant and procedures included - were not described.

The authors mentioned that they developed a systematic review. However, in line 78, they mentioned their results: "Then in more detail, we present results from a scoping review for TASF and health during vulnerable periods of the life course: pregnancy and lactation; childhood; and older adults", which is a confusing statement.

RESPONSE 3

Thank for you these suggestions.  As noted previously, while we aspired to a high quality, comprehensive review (e.g., applying approaches such as PICO/PECO and standards from PRISMA), this was ultimately a narrative review.  Nonetheless, we have integrated several of these details into the Methods section suggested by the reviewer.  We appreciate the advice.

COMMENT 4

The methods section should mention comments that are part of the special consideration (lines 471-475).

4 - results and conclusion

It must be revised according to references 29 and 30, with the study limitation mentioned before the conclusion.

RESPONSE 4

We added narrative to the Methods section to describe the two topics for special consideration.  As well, the study limitations have been moved to come before the conclusion.  Thank you for these comments.

COMMENT 5

5 - Supplementary material should include the registration and protocol and the availability of the data, code and other material.

RESPONSE 5

Please see responses above.  As this was considered a narrative review, we do not have a registration or protocol to include.  That said, as per this reviewer’s comments, we have added more details to the Methods section to more clearly delineate the procedures followed. 

Reviewer 3 Report

Comments and Suggestions for Authors

The paper deals with a relevant research topic: considering nutritional requirements at different life stages.

Animal sources are under scrutiny for environmental concerns and health outcomes. Therefore, it is crucial to state their actual role in human nutrition.

The study design seems suitable, but the development of the manuscript does not allow appreciation of the work done. I suggest to reorganize the text according to the following considerations.

Remarks.

Methods are described in the second part, but a specific section is required.

Table 1 reports the results of the systematic research in the literature, but subsequent paragraphs do not refer to this table and cite further literature. This hides the crucial results, if any.

I suggest stressing the results by commenting on Table 1 and transferring the text related to other articles to a discussion section. The latter should include the text currently in the “conclusions” section.

Minor editing.

Did the Authors record the search protocol on the PROSPERO platform?

Please describe the meaning of the arrows in Table 1. 

Author Response

We greatly appreciate the reviewer’s recognition of the extensive work on this review and manuscript.  The remarks were very helpful towards better communicating the findings.  Thank you so much.

The paper deals with a relevant research topic: considering nutritional requirements at different life stages.

Animal sources are under scrutiny for environmental concerns and health outcomes. Therefore, it is crucial to state their actual role in human nutrition.

The study design seems suitable, but the development of the manuscript does not allow appreciation of the work done. I suggest to reorganize the text according to the following considerations.

Remarks.

COMMENT 1

Methods are described in the second part, but a specific section is required.

RESPONSE 1

Thank you for this comment. The paper has been reformatted to create a clear methods section.

COMMENT 2

Table 1 reports the results of the systematic research in the literature, but subsequent paragraphs do not refer to this table and cite further literature. This hides the crucial results, if any.

RESPONSE 2

Thank you for this comment.  We have now inserted citations and importantly, added narrative describing the findings from the table at the outset of the Results section.  It was helpful to revisit the table and summarize findings by life course phase and TASF.

COMMENT 3

I suggest stressing the results by commenting on Table 1 and transferring the text related to other articles to a discussion section. The latter should include the text currently in the “conclusions” section.

RESPONSE 3

We have added narrative to the findings to elaborate on the summarized evidence base in Table 1, further highlighting our findings as recommended by the reviewer.  The Discussion section, specifically the Conclusions section, has also been reorganized and revised.  

Minor editing.

COMMENT 4

Did the Authors record the search protocol on the PROSPERO platform?

RESPONSE 4

We have revised the manuscript to better reflect the nature of the review, considered a narrative review.  Although we maintained high standards and followed protocols comparative to a systematic review, the study was not registered on the PROSPERO platform.  Thank you for the suggestion in any case. 

COMMENT 5

Please describe the meaning of the arrows in Table 1. 

RESPONSE 5

This has been completed in the footnote of the table, also requested by reviewer 1.

Round 2

Reviewer 3 Report

Comments and Suggestions for Authors

I congratulate the Authors. The text has been greatly enhanced, and it is now easier to understand the importance of their work.

I have a couple of further suggestions for completing the ameliorant. 

Table 1 contains the results of the systematic literature review. I would move this to the results section and comment on it per se.

It would be interesting to explain the content of the table that represents the reader's leading interest: factors affecting health positively by increasing and decreasing and factors affecting health negatively by increasing and decreasing.  

All the subsequent sections (3.1-3.4 and the whole 4) report a point-by-point discussion where further literature is cited. That is correct as a discussion section. 

In this part, a citation of the FAO database on national dietary guidelines (https://www.fao.org/nutrition/education/food-based-dietary-guidelines) regarding the consideration of life course stages is opportune. In lines 73-78, the Authors explain the structure of their publication plan. The present paper seems to be the second one, but the first one is not cited, while the third one should be related to the analysis of TASF's role in the different guidelines as further research. Please add some information in the introduction and a sentence in the discussion section.

The conclusions section reports discussion points. I suggest the Authors include this in the discussion and draw synthetic conclusions, such as TASF is an essential part of a healthy diet, especially at a specific life stage, although in the right proportion as indicated in the dietary guidelines. (This is just an example to give an idea of what I mean by the synthetic conclusion)

Author Response

Manuscript ID: nutrients-3161525

Type of manuscript: Review

Title: Terrestrial animal source foods and health outcomes for those with special nutrient needs in the life course 

REVIEWER 3

I congratulate the Authors. The text has been greatly enhanced, and it is now easier to understand the importance of their work.

I have a couple of further suggestions for completing the ameliorant. 

COMMENT 1.

Table 1 contains the results of the systematic literature review. I would move this to the results section and comment on it per se.

It would be interesting to explain the content of the table that represents the reader's leading interest: factors affecting health positively by increasing and decreasing and factors affecting health negatively by increasing and decreasing.  

RESPONSE 1

Thank you for this comment. Table 1 has been moved to the results section after the introductory paragraph for clarity.  In response to this comment, we embellished and added more direct statements about the positive and negative findings to the narrative describing Table 1, added in the first round of revise and resubmit.

COMMENT 2

All the subsequent sections (3.1-3.4 and the whole 4) report a point-by-point discussion where further literature is cited. That is correct as a discussion section. 

RESPONSE 2

We appreciate this suggestion to move the specified sections into the Discussion.  However, as a narrative review, we would prefer to retain these sections in the Results section to provide greater detail to the summarized findings of Table 1 and include further literature that would be included in a narrative review.  

COMMENT 3

In this part, a citation of the FAO database on national dietary guidelines (https://www.fao.org/nutrition/education/food-based-dietary-guidelines) regarding the consideration of life course stages is opportune. In lines 73-78, the Authors explain the structure of their publication plan.

The present paper seems to be the second one, but the first one is not cited, while the third one should be related to the analysis of TASF's role in the different guidelines as further research. Please add some information in the introduction and a sentence in the discussion section.

RESPONSE 3

Thank you for this comment. A citation of the FAO database on national dietary guidelines has been added in this section after its mention.  In response to the reviewer, we added additional information to further explain the three-paper series, specifying that the present paper is paper #2 and agree with the reviewer that the third paper will include an analysis of TASF role in dietary guidelines as further research.  This narrative was added to both the Introduction and Discussion sections.

COMMENT 4

The conclusions section reports discussion points. I suggest the Authors include this in the discussion and draw synthetic conclusions, such as TASF is an essential part of a healthy diet, especially at a specific life stage, although in the right proportion as indicated in the dietary guidelines. (This is just an example to give an idea of what I mean by the synthetic conclusion)

RESPONSE 4

We really appreciated the reviewer suggestions in this comment and have reworked the Conclusion section, now labeled “Discussion and synthetic conclusions.”